# Seafood Waste as Attractive Source of Chitin and Chitosan Production and Their Applications

**DOI:** 10.3390/ijms21124290

**Published:** 2020-06-16

**Authors:** Vanessa P. Santos, Nathália S. S. Marques, Patrícia C. S. V. Maia, Marcos Antonio Barbosa de Lima, Luciana de Oliveira Franco, Galba Maria de Campos-Takaki

**Affiliations:** 1Federal Rural University of Pernambuco, Recife 52171-900, Pernambuco, Brazil; nessapimentel4519@hotmail.com (V.P.S.); nathalia13@hotmail.com (N.S.S.M.); patriciacvsm@gmail.com (P.C.S.V.M.); 2Department of Microbiology, Federal Rural University of Pernambuco, Recife 52171-900, Pernambuco, Brazil; mablima33@yahoo.com.br (M.A.B.d.L.); lucianafranco39@gmail.com (L.d.O.F.); 3Research Center for Environmental Sciences and Biotechnology, Catholic University Pernambuco, Recife 50050-590, Pernambuco, Brazil

**Keywords:** polysaccharide, biopolymer, shrimp waste, chitosan blend, biotechnology

## Abstract

Chitosan is a cationic polymer obtained by deacetylation of chitin, found abundantly in crustacean, insect, arthropod exoskeletons, and molluscs. The process of obtaining chitin by the chemical extraction method comprises the steps of deproteinization, demineralization, and discoloration. To obtain chitosan, the deacetylation of chitin is necessary. These polymers can also be extracted through the biological extraction method involving the use of microorganisms. Chitosan has biodegradable and biocompatible properties, being applied in the pharmaceutical, cosmetic, food, biomedical, chemical, and textile industries. Chitosan and its derivatives may be used in the form of gels, beads, membranes, films, and sponges, depending on their application. Polymer blending can also be performed to improve the mechanical properties of the bioproduct. This review aims to provide the latest information on existing methods for chitin and chitosan recovery from marine waste as well as their applications.

## 1. Introduction

The reuse of waste from the fishing industry is not a common practice, and large percentage of biomass of waste is discarded directly into the environment without previous treatment [1]. Any material that is not used during its production or consumption process due to technology or market limitations, which can cause damage to the environment when not properly managed, is considered waste [2,3]. However, this type of waste is a source of raw material with high benefits and can be used to produce biocompounds [4].

New processing and management techniques for these wastes are needed in order to generate quality co-products and reduce environmental impacts, therefore, we would have greater job creation and sustainable development of the fishing industry [5,6].

The seafood industry annually generates about 106 tons of waste, most of which is destined for composting or to be converted into low value-added products such as animal feed and fertilizers [7]. In this context, approximately 2000 tons of chitosan is produced annually, whose main source of extraction is from shrimp and crab shell residues [8].

Chitin is a natural polysaccharide found in fungi cell walls as well as in crustacean and insect exoskeletons and is the second most abundant biopolymer in nature after cellulose. Chitin is converted to chitosan by deacetylation that involves the removal of the acetyl group [9,10].

Chitosan is widely used as a biomaterial because it has biological properties of biocompatibility, biodegradability, and atoxicity. It can be used as a therapeutic agent because it has antibacterial and antifungal characteristics, making it interesting for applications in agriculture, medicine, the environment, and the food, cosmetic, and textile industries [11,12,13].

Therefore, the purpose of this review is to provide the latest information on existing methods for chitin and chitosan recovery from marine waste as well as their applications.

## 2. Chitin and Chitosan History 

The research on chitin isolation and characterization began in 1811 by the studies of the French chemist Henri Braconnot, in which some fungal species were subjected to an aqueous alkali treatment, which allowed the extraction of the fungine, named by him. In 1843, Lassaigne conducted research from exoskeletons of the species *Bombyx mori* (silkworm), in which he demonstrated the presence of nitrogen in the structure of chitin. Chitosan was discovered in 1859 by treating chitin with heated potassium hydroxide. In 1878, Ledderhose pointed out that chitin has compounds such as glycosamine and acetic acid; however, only in 1894, Gilson confirmed the presence of glycosamine units. Still, in 1894, the German Felix Hoppe-Seyler named the compound chitosan. It was in 1950 that the chemical structure of chitosan was determined [14,15].

The first reports of chitosan production appeared in 1970 in Japan and the United States. By 1986, Japan already had fifteen industries producing chitin and chitosan commercially. Japan and the United States are world leaders in chitosan production, standing out in the research of this polysaccharide in the multitude of applications of chitosan, being economically attractive and profitable [16,17].

In 1983, the first studies with fungal chitosan were carried out in Brazil, confirming the presence of chitosan in the cell walls of fungi belonging to the class Zygomycetes and Mucorales [18]. The first records of chitosan production and marketing were done by Craveiro, Craveiro, and Queiroz (1999) [19]. Most Brazilian researches are still on a bench scale, with few patent filings being observed; however, they are extremely important because of their biotechnological importance and the potential of research groups operating in the Brazilian Northeast [20].

## 3. Occurrence of Chitin in Nature

Chitin is a natural polymer that has a highly organized crystalline structure that is nitrogenous, white, and hard, having a low chemical reactivity. It is the second most abundant polysaccharide in nature, second only to cellulose. It is insoluble in water and organic solvents, presenting, after purification, as a yellowish powder. It has a high molecular weight and is chemically composed of *N*-acetyl-2-amino-2-deoxy-D-glucose units joined together by glycosidic bonds β (1 → 4) (Figure 1), forming a linear chain with some of the deacetylated monomer units [21,22,23].

Chitin is widely distributed in nature and is the main element of the marine invertebrate exoskeleton and can be found in the structure of insects, arthropods, and molluscs [24,25,26]. Figure 2 presents the main sources of chitin production and its extraction.

Chitin polymorphism can be visualized using X-ray diffraction, where three crystalline structures are observed, α, β, and γ, which differ by the number of chains per cell, degree of hydration, and unit size. The α-chitin is the most abundant form, being found in arthropod exoskeletons, where the dispositions of the polymeric chains are antiparallel, which favors the existence of numerous inter- and intra-chain hydrogen bonds that result in a densely packed material (Figure 3). In β-chitin, the disposition is parallel and they are found in animals that show flexibility and resistance, such as squids. The γ-chitin displays a mixture of both positions [27,28].

Chitin derivatives have high economic value due to their biological activities and applications, being biodegradable and biocompatible polymers as well as produced by renewable natural sources. Harnessing the by-products of crustacean processing is a profitable activity because of their richness in high value-added compounds [29,30].

## 4. Biosynthesis of Chitin

Chitin occurs in complexes strongly linked with other substances in the cuticles of crustaceans, and some portions of polypeptides are linked to a small number of amino groups [31].

Generally, the carbon source used for chitin synthesis is glucose, starting the process of glycogen catalysis by the enzyme phosphorylase and being converted into glucose-1-P. In the presence of phosphomutase, glucose-6-P is formed and further converted to fructose-6-P by hexokinase. Intracellular fructose-6-P is converted to glucosamine-6-P via aminotransferase using L-glutamine. Then, glucosamine-6-P is converted to N-acetylglucosamine-6-P via N-acetyltransferase using acetyl co-A as a substrate. The phosphate group on it is changed from the 6-P to the 1-P position by phosphoacetylglucosamine mutase. Subsequently, N-acetylglucosamine-1-P is converted to UDP-N- acetylglucosamine via pyrophosphorylase using triphosphate as the cosubstrate. Chitin is formed from UDP-N-acetylglucosamine in the presence of the enzyme chitin synthase (Figure 4) [32,33,34]. The chitin deacetylation reaction results in chitosan [35,36].

## 5. Chitin Isolation from Natural Resources 

Seafood is a major source of animal protein in many countries, however, besides the edible part, these raw materials have an inedible one [37]. A significant part of the environmental contamination is caused by the wastes from fishing industries, which triggers an environmental problem due to their unpleasant odor, attracting and stimulating the proliferation of insects. They can also be harmful to human health when disposed of without any kind of previous treatment [38,39].

The effluents resulting from the fishing industry, if released without previous treatment in the environment, cause physical and chemical changes in water bodies, which may cause mortality of aquatic animals and impact the local microfauna and microflora, given that this residue is characterized by its high concentrations of nitrogen, phosphorus, organic carbon, suspended solids, and oxygen [1].

Seafood waste is a potential source of raw material for chitin extraction [40]. However, due to their origin from natural resources and their chemical and physical variability, the properties of chitin and chitosan can have a direct impact on their applications [41]. Characteristics that may be related to the extraction process are molecular weight, degree of deacetylation, degree of purity, viscosity, and crystallinity [42].

### 5.1. Chemical Extraction

This type of extraction consists of the use of a strong alkaline solution, such as hydrolysis with sodium hydroxide at high temperatures and concentrations, causing the breakdown of polymeric chains, and establishing a high degree of chitosan deacetylation [43].

The chemical extraction method is composed of three basic steps (Figure 5), an alkaline solution deproteinization, an acid solution demineralization, and a discoloration. It is noteworthy that all these steps are directly related to the physicochemical properties of the chitin obtained [44,45]. The source for chitin extraction is subjected to washing, drying, and grinding of powder particles [46].

This type of conventional extraction can cause problems in the disposal of waste generated, which is necessary for the neutralization and detoxification of wastewater [47].

#### 5.1.1. Chemical Deproteinization

The deproteinization involves the disruption of chemical bonds between proteins and chitin, requiring chemicals to depolymerize the biopolymer [48], whose removal of the associated proteins is an essential step in the polysaccharide purification process [49,50].

Conventional extraction of chitin from marine waste through the deproteinization involves the use of bases and strong acids at high temperatures, which demands high energy consumption and generates effluents with high chemical concentrations, requiring appropriate treatment for their neutralization [51].

The use of strong acids and bases during the chitin extraction process leads to an increase in the cost of materials involved in the process, as well as a low-purity end product [52].

#### 5.1.2. Chemical Demineralization

Demineralization is a process for the removal of minerals, especially calcium carbonate, using strong acids [53]. The most commonly used acids in this treatment process are sulfuric acid, hydrochloric acid, acetic acid, nitric acid, and formic acid [54,55,56].

Demineralization occurs through the decomposition of calcium carbonate in calcium chloride, with the release of carbon dioxide [57], as shown in the reaction:(1)2HCl+CaCO3 →CaCl2+H2O+CO2↑

#### 5.1.3. Discoloration 

This is an additional step during the extraction process and is performed if you wish to obtain a colorless product as it aims to eliminate astaxanthin and β-carotene pigments when they are present in the extraction source. It uses organic or inorganic solvents such as acetone, sodium hypochlorite, and hydrogen peroxide [42].

### 5.2. Biological Extraction

The biological extraction method involves the use of microorganisms that produce enzymes and organic acids at a relatively low cost, with a cleaner and greener process, favoring the production of quality chitin [58,59].

The biological extraction process has been made more attractive by obtaining high-quality products, with the cost of production being affordable and not generating high concentration chemical effluents, as mentioned in the chemical process [60].

Biological methods often used for chitin extraction are enzymatic deproteinization and fermentation using microorganisms [61,62].

#### 5.2.1. Enzymatic Deproteinization

Enzymatic deproteinization of the waste from the fishing industry to obtain hydrolyzed protein is a method based on the addition of enzymes for protein fragmentation, having the advantage of not producing environmental degradation products [51,62].

Proteases are of utmost importance for protein removal during chitin extraction from fishing industry waste [63]. The proteases involved in the protein removal process from seafood residues are papain, trypsin, pepsin, alkalase, and pancreatin [64,65].

#### 5.2.2. Fermentation

Hydrolyzed proteins can be obtained by proteolytic enzymes produced by the lactic acid bacteria activated due to a low pH in the medium. The advantage of this process is that it allows the recovery of value-added by-products such as proteins, enzymes, and pigments that can be applied, for example, in the food industry [66].

The efficiency of fermentation through microorganisms depends directly on the amount of inoculum, glucose concentration in the medium, the pH during the culture, and fermentation time. This type of extraction using microorganisms is a tendency in biotechnology and bioremediation researches [67].

Fermentation can be performed using protease-producing bacteria such as *Bacillus subtilis*, *Pseudomonas aeruginosa*, *Pseudomonas maltophilia*, and *Serratia marcescens* [68,69,70].

## 6. Chitin to Chitosan Conversion 

Chitosan is a polysaccharide obtained by chitin deacetylation reaction through alkaline hydrolysis and subsequent treatment with acid solutions, consisting of 2-amino-2-deoxy-D-glycopyranose units joined by glycosidic bonds β (1 → 4) (Figure 5). However, the polymers differ in relative proportion and solubility of these units. They can function as an ion exchange resin for being soluble in organic acids and diluted minerals; nevertheless, their precipitation occurs with a pH value above 6.0 [71,72,73].

Chitosan are all chitin derivatives having a degree of deacetylation of 50% or more. The relative proportions of these units generate distinct structural characteristics, such as the degree of deacetylation and molecular weight, whose structural characteristics are related to the physicochemical and biological properties of the polymer [30,74,75].

Chitosan has the characteristic of solubility in acidic media because of the free amino groups being protonated (NH3 +), where the precipitation tendency increases from the moment the pH approaches 6.0, regarding the increase of -NH2 clusters in the molecular structure. Thus, amino groups make it possible for them to bind to negatively charged materials, such as other polysaccharides, enzymes, and cells, being insoluble in water, concentrated acids, acetone, and alcohol [76,77,78].

In recent years, many studies have been performed on chitosan and have shown a close dependence relationship between the structural and morphological characteristics of chitin, chitosan and their derivatives, their properties, and potential applications [79].

This polymer can be found in nature in small quantities in the cell walls of some fungi (*Zygomycetes*), or it can be obtained by alkaline hydrolysis of chitin from crustacean and exoskeletons of arthropods [80]. Crustacean shells contain 15% to 20% of chitin, 25% to 40% of protein, and 40% to 55% of calcium carbonate; the latter is responsible for crustacean rigidity [81].

## 7. Chitin and Chitosan Blend with Other Polymers 

Polymer blending is defined as the homogeneous mixture of two or more different polymer species. Many blends are produced for financial reasons to reduce the costs of a technical application. However, the blend may lead to better properties of the product obtained [82].

Chitin and chitosan can be combined with poly (vinyl alcohol), alginate, collagen, cellulose acetate, among others to improve their mechanical properties [83,84,85,86]. Because chitosan is soluble in aqueous acid solutions, it can take different shapes, such as particles, films, sponges, membranes, gels, fibers, including others [87].

## 8. Chitin and Chitosan Applications 

New technological approaches are needed to improve human lives and the environment. What makes chitosan a material of industrial interest are its broad spectrum of properties, such as water insolubility, cationic biopolymer behavior, positive global charge in biological pH, and easily capable of forming gels. These properties make it interesting for agricultural applications, medicine, environment, and food, as indicated in Figure 6 [88,89].

### 8.1. Active Ingredient Carrier

Nanotechnology has applications in many areas, such as engineering, medicine, pharmaceuticals, and agriculture, revolutionizing various processes and products [90]. Chitosan is a polymer of great industrial and biotechnological interest because of its abundant extraction source, being biocompatible and positively charged, defining it as a potential material for the active ingredient delivery system [91].

Chitin and chitosan-based nanomaterials can be used as carriers of cosmetic ingredients, such as chitin nanofibrils face masks capable of releasing active ingredients at different doses and time, and can be used as antibacterial, anti-inflammatory, sunscreen, anti-aging cosmetics depending on the active ingredient selected [92].

Nanotechnology can also be applied in agriculture, favoring agroindustry to be greener. Nanoparticles can increase the effectiveness of agrochemicals, resulting in lower doses and fewer applications, as well as reducing the risk of environmental contamination and promoting effective pest control in agriculture [93,94].

### 8.2. Tissue Engineering

The engineering of artificial tissues represents major advances in the biomedical field as it assists in reconstructive processes, favoring improvement in human life quality [95]. It involves the regeneration of lost or damaged tissues using biomaterials associated with cell or growth factors [96].

An important requirement for scaffolding is to have an interconnected structure with high porosity to ensure proper nutrient penetration and diffusion into cells [97].

Characteristics for choosing a biomaterial for tissue engineering are: presence of interconnected pores, controlled biodegradability, modifiable chemical surface, mechanical properties similar to the site of implantation, insignificant toxicity, and ease of obtaining desirable shapes and sizes [98].

### 8.3. Active Pharmaceutical Applications

Several polymers have been used in the production of mucoadhesive delivery systems; nevertheless, chitosan and its derivatives are the most broadly used due to characteristics of atoxicity, biocompatibility, antimicrobial activity, and adequate permeation [99].

Chitosan-based nanoparticles are widely used as devices for drug administration because they have useful features as a drug-loading vehicle. It has a biological property of mucoadhesiveness, implying the transient opening of epithelial junctions for drug entry [100].

Chitosan is a natural polyaminosaccharide with a non-toxic, non-allergenic, biocompatible, and biodegradable characteristic, and derivatives from chitosan are reported as anticoagulants. The literature described that the chitosan has a similar close structure like heparin, and based on this feature, many molecules of chitosan derivatives have been synthesized [101,102,103,104,105].

Anticoagulants are clinically used in different medical conditions like thrombosis and have the maximum annual growth rate among the top ten treatment areas. The developed compounds exhibited a faster onset of action and potency than nicoumalone after one hour of the drug administration. The sulphated N-alkyl derivatives of chitosan were suggested as the more potent anticoagulants than sulfated quaternary derivatives/sulfated chitosan [106,107]. 

In this sense, the most prominent commercial application of chitosan is its use as a hemostatic functional system. Therefore, chitosan-based wound dressings are available on the market for clinical use as products HemCon® Bandage and ChitoFlex wound dressings (HemCon Medical Technologies, UK), as well as CELOX™ (Medtrade Products, England); all products are FDA approved (http://www.hemcon.com and http://www.celoxmedical.com, respectively) [108].

### 8.4. Antimicrobial Agent 

Many investigations reported the chitosan antimicrobial activity, but the mechanism has not yet been fully elucidated and studies are fundamental in the search for clarification of the potential of chitosan. One of the most studied properties of chitosan is its antimicrobial activity, related to the ability of its positively charged amino groupings to bind to the surface of the bacterial wall or the plasma membrane because they have negative charges. Thus, there is a change in cell permeability, favoring the flow of ions and proteins from the cytoplasm into the extracellular space and causing cell death [109]. Higher degree of acetylation, higher molecular weight of chitosan, and the antibacterial activity mediates the changes in cell permeability and blocks the transport of the bacteria [110,111,112]. Lower degree of acetylation of chitosan and lower pH favor antibacterial activity; however, the reduction of the molecular weight and the activity is toward Gram-negative bacteria, as well as the molecular weight and degree of acetylation influenced the antifungal activity with various fungi [113].

Other factors related to chitosan antimicrobial activity are the absorbing property of metal ions and their ease of penetrating the cell wall and binding to DNA, inhibiting messenger RNA synthesis, and low molecular weight of chitosan molecule induced inhibition of DNA transcription and mRNA synthesis in *E. coli* [110,111,112,113,114,115].

Therefore, the antimicrobial activity of chitosan and its derivatives is widely explored for the production of self-preserving materials through food protection and packaging. Chitosan films have a large application in food packaging materials, forming a protective antimicrobial barrier and preserving the nutritional quality of foods [116,117].

### 8.5. Water Treatment

Water and the range of services generated by this depleted natural resource contribute to poverty reduction, economic growth, as well as social and environmental sustainability, contributing to improvements in social welfare [118].

Wastewater from the food, textile, vegetable oil processing, oil production, and domestic sewage companies is a major source of pollution, given that it contains various organic compounds and is not properly treated before being discharged into the effluents [119].

Adsorption is the adhesion or fixation of molecules or electrostatic bonding of a fluid to a solid surface, enabling the elimination of compounds, metal ions, or other materials using an inactive sorbent of biological origin or natural products, by means of attractive forces between the material removed and the biosorbent [120]. The literature cites some adsorbents that are effective in removing toxic metal ions and are environmentally friendly, such as chitin, chitosan, cellulose, and guarana [121]. Chitosan is used to remove oils, greases, and heavy metals [122,123]. 

The increase of the degree of chitosan deacetylation is related to a greater number of amino groups, which are the main sorption centers, with being the degree of deacetylation and pH the main factors that affect the absorption capacity of chitosan [124].

### 8.6. Chitosan Applications in Food Technology

Chitosan biopolymer is biocompatible, nonantigenic, nontoxic, and biofunctional molecule and has attracted notable attention as a potential food preservative of natural origin [125,126,127]. Chitosan from shrimp isolation was preconized as GRAS (Generally Recognized As Safe) based on the scientific procedures for multiple technical effects in several food categories (GRAS Notice No. GRN 000443) [108,128]. 

Applications of chitosan were investigated for extension of shelf life of bread by retarding starch retrogradation and/or by inhibiting microbial growth have been observed. The authors evaluated chitosan molecule with 493 kDa coating on shelf life of baguette surface using 0.5%, 1.0%, or 1.5% chitosan diluted in 1.0% acetic acid. The results indicated barrier properties of chitosan to baguette-coated 1% chitosan, less weight loss, hardness, and retrogradation compared with the control during storage for 36 h at 25 °C [129,130]. 

Chitosan obtained by extraction from a bio-waste product using many energy-efficient methods. Chitosan is much cheaper as compared to other biopolymers. Nevertheless, the exceptional properties of chitosan make it a relatively stronger candidate for food packaging applications. The most popular and most economical way for production of chitosan is from deacetylation process of chitin. However, it is also possible to obtain chitosan directly from some fungi cell walls [131,132] and other organisms [133,134]. Chitosan film has been widely used to extend the shelf life of food, with addition of Ca^2+^ ions changes the permeation rate of CO_2_ and O_2_ through the chitosan membranes and increases the useful life of the fruits [17] Raw materials to prepare a series of films of chitosan and glycerol formulation showed strawberry preservation [135]. Coating with chitosan films by immersion in a 1% polysaccharide solution containing 0.1% Ca^2+^ prevents changes in the sensory properties of vegetables. In addition, chitosan is also useful in the production of paper for food packaging coated with it, which acts as an inhibitor of microbial growth [136]. Films formed by chitosan and polyvinyl alcohol with lignin nanoparticles are characterized by increased strength compared to films formed by individual components, antibacterial action against gram-negative microorganisms, and the synergistic antioxidant effect of chitosan and lignin [137].

## 9. Conclusions and Future Trends

As a major by-product of the seafood waste, a massive amount of crustacean shell waste is generated each year that can be used to produce value-added chitin, which can be converted to chitosan using a relatively simple deacetylation process. As the bio-waste product using many energy-efficient methods, chitosan is much cheaper as compared to other biopolymers. In the present review, chitin and chitosan has been presented as ideal renewable agents for native form or upgraded and incorporated with antimicrobial particles and natural compounds with multifunctional applications. Demands for alternative materials in various fields of biotechnology and industry are driven by technological advancement, favoring the increasing use of biopolymers and having chitosan as the most abundant and renewable polysaccharide, thus attracting more attention from researchers.

Chitin can be easily obtained from marine animals, crustacean residues, insects, and microorganisms. Chitosan is obtained by deacetylation of chitin, presenting antibacterial and antifungal properties, biocompatibility, mucoadhesivity, atoxicity, among others. Since previous decades, chitosan has been very important in several industrial applications, including biomedicine, textile, food, pharmaceutical, and cosmetic industries. For future applications, chitosan-based materials can be used as advanced composites or fibers, having a promising utility for accelerating tissue repair and wound healing processes for the pharmaceutical and biomedical industries. In addition, chitosan has become attractive to boost studies involving tissue engineering.

Therefore, this polymer is very attractive for application in several areas due to its characteristics, giving this polysaccharide a very promising future as a biomaterial. These include development of new smart future trends to biomaterials as promising wound healing effects and therapeutic molecules that are released at the same time as the microbial growth.

## Figures and Tables

**Figure 1 ijms-21-04290-f001:**
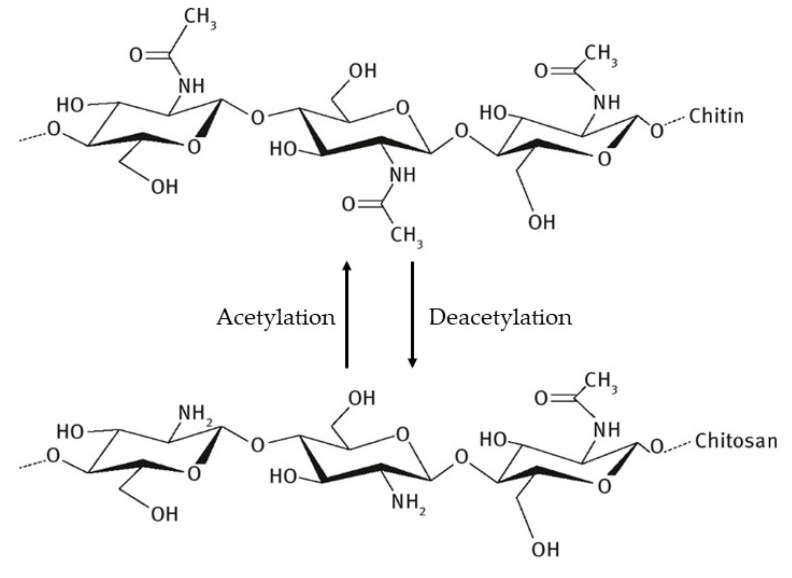
Chemical structures of chitin and chitosan.

**Figure 2 ijms-21-04290-f002:**
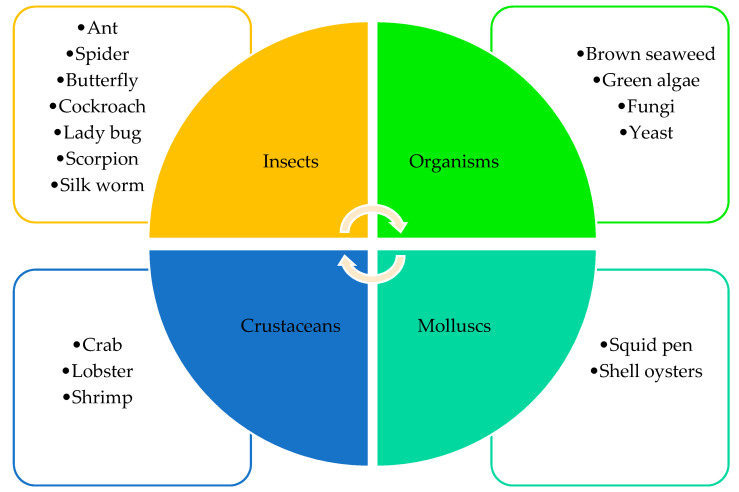
Sources of chitin production.

**Figure 3 ijms-21-04290-f003:**
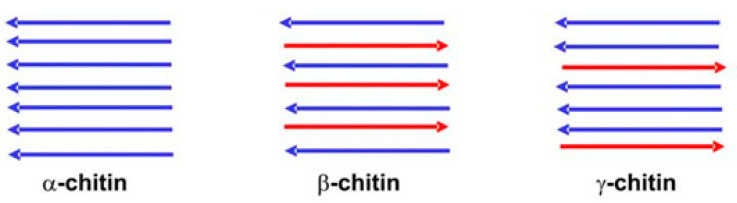
Polymorphic structures of chitin.

**Figure 4 ijms-21-04290-f004:**
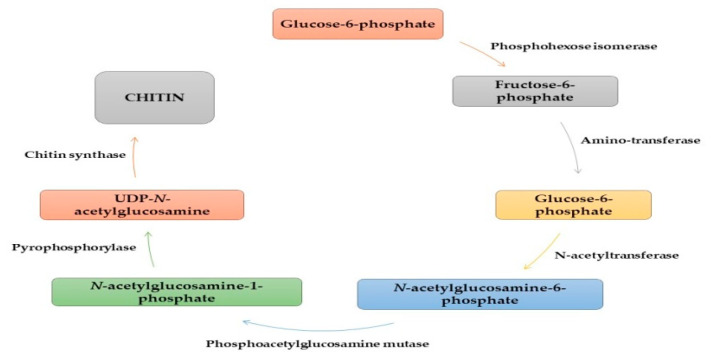
General scheme of chitin biosynthesis in biological systems.

**Figure 5 ijms-21-04290-f005:**
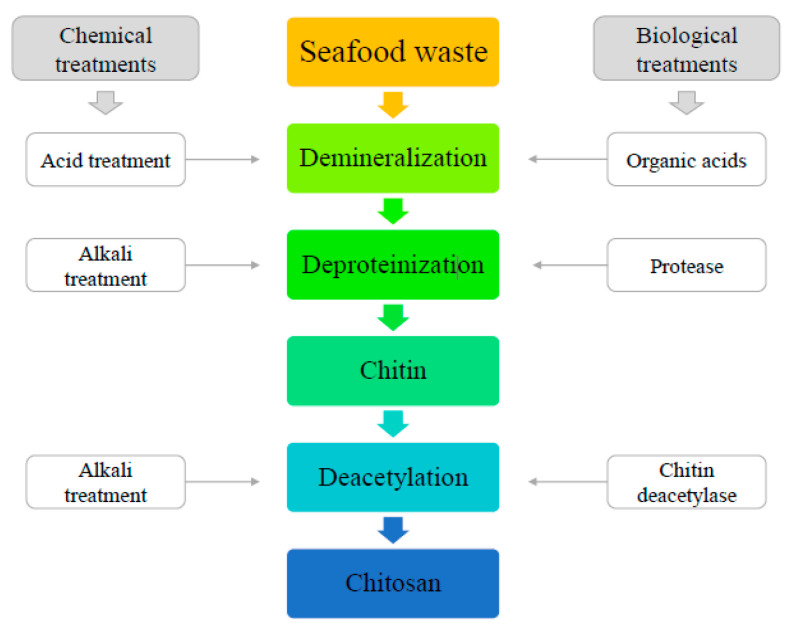
Chitin and chitosan production by chemical and biological treatments.

**Figure 6 ijms-21-04290-f006:**
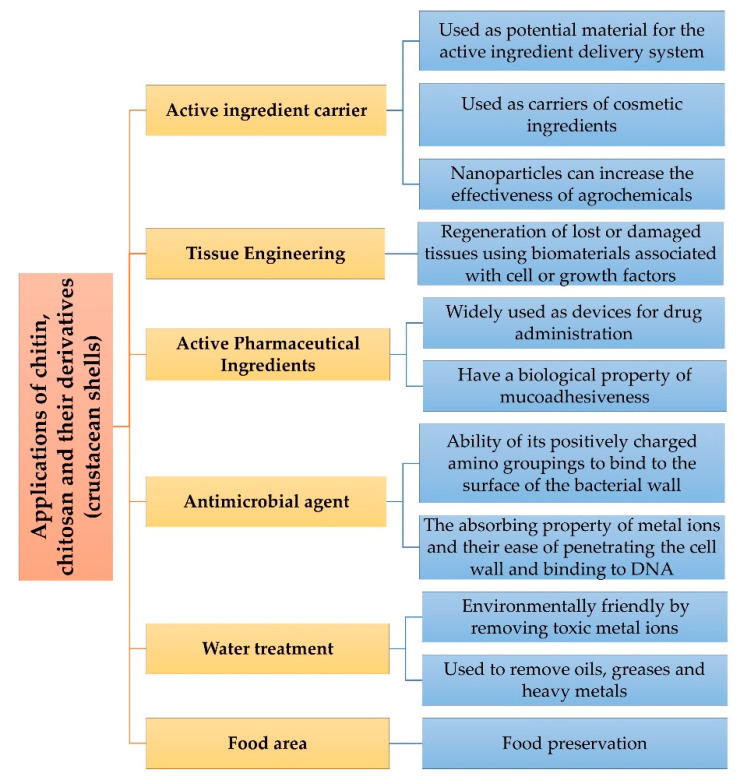
Flowchart of the summary of the main applications of chitin/chitosan.

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
