# Peer review of "Seafood Waste as Attractive Source of Chitin and Chitosan Production and Their Applications"

_ijms, 2020, doi:10.3390/ijms21124290_

Round 1

Reviewer 1 Report

Respected Authors,

The manuscript has collected and processed a significant data related to the production and use of chitin and chitosan. In the present Manuscript, the authors provide an excellent scientific review on the analyzed subject, with a clear scientific discussion and a great contribution to the scientific field. The Manuscript is very clearly written and very understandable.

Each topic in the manuscript provides meaningful information and systematically describes the current state of the art within the area under study. Relevant literature is used in the paper, and the manuscript is supported by the previous work of the authors themselves.

My only suggestions are for the following items:

Line 124: Please replace the word ″social″ with the word ″environmental″.

Line 204: Please align figure quotation - upper or lower case (for example, line 141 and line 204 are not aligned, please chose 'Figure' or 'figure').

The recommendation is that the Manuscript can completely meet the scientific needs and demands, and can be accepted for publication.

Author Response

REEREE 1  -

 Comments and Suggestions for Authors

Respected Authors,

The manuscript has collected and processed a significant data related to the production and use of chitin and chitosan. In the present Manuscript, the authors provide an excellent scientific review on the analyzed subject, with a clear scientific discussion and a great contribution to the scientific field. The Manuscript is very clearly written and very understandable.

Each topic in the manuscript provides meaningful information and systematically describes the current state of the art within the area under study. Relevant literature is used in the paper, and the manuscript is supported by the previous work of the authors themselves.

My only suggestions are for the following items:

Line 124: Please replace the word ″social″ with the word ″environmental″.

We have done.

Line 204: Please align figure quotation - upper or lower case (for example, line 141 and line 204 are not aligned, please chose 'Figure' or 'figure').

We have done.

The recommendation is that the Manuscript can completely meet the scientific needs and demands, and can be accepted for publication.

Reviewer 2 Report

I recommend author to revise this manuscript.

  1. Author must include valid molecular mechanism including data to validate their claims.
  2. Author must include proper biosynthesis processes in figure style with text in section 4.
  3. Application part must have self explanatory figures just tissue engineer approach in rectangle are not enough to explain.
  4.  Also include mechanisms in application parts.
  5. Conclusion must include proper future prospective in detail

Author Response

REFEREE 2

Comments and Suggestions for Authors

I recommend author to revise this manuscript.

  1. Author must include valid molecular mechanism including data to validate their claims.

We have done.

  1. Author must include proper biosynthesis processes in figure style with text in section 4.

We have done and included  figure-shaped biosynthesis process (figure 4) was included in the line 111.

  1. Application part must have self explanatory figures just tissue engineer approach in rectangle are not enough to explain.

We have done . The rectangles on the use of chitosan in engineering tissue have been removed, as it was added all applications of chitosan in a single figure in the section "Chitin and Chitosan applications.

  1.  Also include mechanisms in application parts.

We have done .  Figure was added in the line 226  representative  of the applications of chitosan and its derivatives.

  1. Conclusion must include proper future prospective in detail

We have done .  

Reviewer 3 Report

The authors did a lot of papers to organize the review paper, it can help some researchers who are in the area. However, I have some suggestions for the authors. 

  1. As we know, Chitosan has "Molecular weight", would the authors can organize the information for the review paper? This information can help some researchers more understanding to use Chitosan or Chitin.
  2. Would the authors organize or explain soluble and insoluble Chitosan for the review paper? If the authors can do it, the review paper may complete. 
  3. The authors mentioned that Chitin or Chitosan be applied in the food also, however,  in line 233-302, I did not read about Chitin and Chitosan applications for food. Would the authors do supplementary explanations or additional materials about the food area for the review paper?

Author Response

Open Review  3

Comments and Suggestions for Authors

The authors did a lot of papers to organize the review paper, it can help some researchers who are in the area. However, I have some suggestions for the authors. 

  1. As we know, Chitosan has "Molecular weight", would the authors can organize the information for the review paper? This information can help some researchers more understanding to use Chitosan or Chitin.

We have done.

  1. Would the authors organize or explain soluble and insoluble Chitosan for the review paper? If the authors can do it, the review paper may complete. 

In section 6 "Chitin to Chitosan conversion", between the lines 199 and 204, it is mentioned in the text about the solubility and insolubility of chitosan.

  1. The authors mentioned that Chitin or Chitosan be applied in the food also, however, in line 233-302, I did not read about Chitin and Chitosan applications for food. Would the authors do supplementary explanations or additional materials about the food area for the review paper?

We have done. A subsection was added on applications of chitosan in the food area. Line 288.

Round 2

Reviewer 2 Report

I recommend to accept this paper as it is revised properly